# A Multi-Task Network Based on Dual-Neck Structure for Autonomous Driving Perception

**DOI:** 10.3390/s24051547

**Published:** 2024-02-28

**Authors:** Guopeng Tan, Chao Wang, Zhihua Li, Yuanbiao Zhang, Ruikai Li

**Affiliations:** 1School of Information & Electrical Engineering, Hebei University of Engineering, Handan 056038, China; 17749954891@163.com (G.T.); lizhihua@hebeu.edu.cn (Z.L.); iszhangyb@foxmail.com (Y.Z.); gealpety@outlook.com (R.L.); 2Hebei Key Laboratory of Security & Protection Information Sensing and Processing, Handan 056038, China

**Keywords:** vehicle detection, drivable area segmentation, lane line segmentation, multi-task learning

## Abstract

A vision-based autonomous driving perception system necessitates the accomplishment of a suite of tasks, including vehicle detection, drivable area segmentation, and lane line segmentation. In light of the limited computational resources available, multi-task learning has emerged as the preeminent methodology for crafting such systems. In this article, we introduce a highly efficient end-to-end multi-task learning model that showcases promising performance on all fronts. Our approach entails the development of a reliable feature extraction network by introducing a feature extraction module called C2SPD. Moreover, to account for the disparities among various tasks, we propose a dual-neck architecture. Finally, we present an optimized design for the decoders of each task. Our model evinces strong performance on the demanding BDD100K dataset, attaining remarkable accuracy (Acc) in vehicle detection and superior precision in drivable area segmentation (mIoU). In addition, this is the first work that can process these three visual perception tasks simultaneously in real time on an embedded device Atlas 200I A2 and maintain excellent accuracy.

## 1. Introduction

The advent of 5G base stations is ushering in an era of 5G-V2X and cloud computing applications for autonomous driving, where intelligent vehicles and smart roads will be the primary manifestations. A robust and dependable driving perception system is crucial for realizing the potential of “smart vehicles”. Although the current driving perception systems primarily rely on a combination of radar and vision algorithms, radar is unable to perform stable classification, detection, and road marking. Hence, a dependable autonomous driving perception system cannot solely rely on radar for perception. Currently, vision algorithms play an indispensable role in constructing a secure and dependable autonomous driving perception system. The provision of crucial information, such as obstacle, lane, and traffic sign data, is an essential requirement for an autonomous driving perception system. To achieve this, the system must possess visual perception capabilities, such as vehicle detection, drivable area segmentation, and lane marking segmentation.

For an autonomous driving perception system, high precision and strong real-time capabilities are essential to ensure timely and accurate decision making for safe vehicle operation. However, computational resources are typically limited in current autonomous driving perception systems. As a result, it is challenging to balance the requirements of precision and real-time performance in a panoramic driving perception system. In the field of computer vision, numerous high-precision and real-time networks have emerged for tasks such as object detection and semantic segmentation. For vehicle detection tasks, algorithms such as Faster R-CNN [1] and the YOLO family [2,3,4,5,6,7] have achieved remarkable performance. Regarding feasible area segmentation, networks such as UNet [8], SegNet [9], and PSPNet [10] are particularly robust. As for the more complex task of lane detection and segmentation, networks such as LaneNet [11], SCNN [12], and LineCNN [13] can satisfy the performance requirements of actual scenes. However, when these three algorithms, designed for different tasks, are implemented within the same autonomous driving perception system, they will encounter issues such as inadequate computational resources and high latency. Furthermore, in practical scenarios, there exists a certain spatial correlation among the three tasks of vehicle detection, drivable area segmentation, and lane detection and segmentation. For instance, vehicles, drivable areas, and lane markings are usually located in the middle and lower parts of the field of view. Vehicles typically border the drivable area, and lane markings typically reside at the edges of the drivable area and border it. This suggests that the networks corresponding to the three tasks have certain similarities in terms of target information extraction and utilization. Therefore, a multi-task learning network is more suitable for deployment in autonomous driving perception systems to accomplish visual tasks such as traffic object detection, drivable area segmentation, and lane detection. There are three specific reasons for this: (1) multiple tasks share one model, saving memory usage and making it easier to deploy in practical scenarios; (2) multiple tasks can generate results in one forward inference pass, improving the real-time performance of the model; (3) the performance of each task can be improved by sharing information.

Currently, the most representative multi-task learning network for autonomous driving scenarios is YOLOP, which consists of an encoder and three different heads and has achieved good performance on the challenging BDD100K dataset [14]. However, due to the requirements of safety and reliability, there is still room for improvement in the detection accuracy of traffic targets and the segmentation accuracy of lane lines. Therefore, this paper proposes a YOLOP-DN multi-task learning network based on a dual-neck structure, which is an improvement on YOLOP. Our proposed network, YOLOP-DN, achieved notable improvements over the baseline model in challenging tasks, including vehicle detection with an mAP value of 78.1, drivable area segmentation with a mIoU value of 92, and lane prediction with an accuracy of 73.8, based on evaluation on the BDD100K dataset. Furthermore, as shown in Figure 1, our driving perception network has achieved impressive results in real-time inference on the Atlas 200I A2.

Our contributions can be summarized in three aspects:(1)We propose a feature extraction backbone network by introducing a feature extraction module called C2SPD. This backbone network achieves better feature extraction without excessively increasing the number of parameters or decreasing the inference speed.(2)We introduce a dual-neck structure. This structure can satisfy the learning requirements of detection and segmentation tasks for different features when fusing features of different resolutions. Moreover, two segmentation tasks share one neck structure, enabling complementary learning between related tasks.(3)We propose an end-to-end multi-task learning network called YOLOP-DN, which is designed for traffic object detection, drivable area segmentation, and lane line detection. The network demonstrates outstanding performance on the BDD100K dataset and has been deployed on the Atlas 200I A2 for practical use.

## 2. Related Work

In this section, we review the solutions for three key perception tasks in autonomous driving: vehicle detection, drivable area segmentation, and lane detection and segmentation. We highlight some of the state-of-the-art networks and provide an overview of the research and application of multi-task learning networks in the context of autonomous driving.

### 2.1. Vehicle Detection

Object detection is one of the most important and challenging tasks in computer vision. In recent years, with the rapid development of deep learning, deep learning models have been widely applied in the field of object detection. Deep learning-based object detection models are generally divided into two categories: single-stage models and two-stage models. Single-stage detectors, such as the YOLO-series and SSD-series [15] algorithms, have higher inference speed, whereas two-stage detectors have higher localization and recognition accuracy, with the Faster R-CNN network being a typical example. Due to the real-time requirements in practical scenarios, single-stage detectors are usually more suitable for real-time vehicle detection. Therefore, in the perception system of autonomous driving, single-stage object detection networks are widely adopted.

At present, YOLOv5 is one of the mainstream single-stage object detectors, with an image inference speed of up to 140 frames per second (i.e., 0.007 s per frame), meeting the real-time detection requirements of video images. Moreover, the network structure is concise, thus being widely applied in many practical tasks, including traffic target and pedestrian detection, traffic sign detection, and so on. At the same time, there have also emerged some improved networks based on YOLOv5 [4], such as YOLOv7 [5] and YOLOv8 [6], which demonstrate stronger detection performance in practical applications.

### 2.2. The Segmentation of Drivable Areas and Lanes

Semantic segmentation is a classic task in the field of computer vision, which aims to achieve pixel-level dense predictions and assign specific semantic labels to each pixel, distinguishing semantically related pixel regions as objects with high-level semantic meaning. In the task of drivable area segmentation, the PSPNet network, based on the pyramid pooling module that integrates global contextual information, effectively performs drivable area segmentation by leveraging global prior information. In addition, networks such as BiFPNet [16], SegNet, and EdgeNet [17] have also achieved excellent performance in the task of drivable area segmentation.

In the task of lane segmentation, the shape features of lane lines are typically elongated, sparse, and fragmented. Therefore, this task often requires more detailed features to achieve effective segmentation. The SCNN network treats the rows or columns of the feature map as layers and uses convolution, non-linear activation, and summation operations to form a spatially deep neural network. This network enables information to propagate between neurons within the same layer, thus enhancing spatial information through inter-layer propagation and maintaining the smoothness and continuity of the lane lines. In addition, Enet-SAD [18] employs a knowledge distillation strategy based on self attention to achieve excellent results in inter-layer information propagation, allowing the network to retain contextual information of the scene in deep features. These two networks demonstrate good performance in lane segmentation tasks.

### 2.3. Multi-Task Learning Network

Currently, the structure of multi-task learning networks mainly consists of an encoder and decoder for each task. The encoder is used to learn generalized features between different tasks to prevent overfitting, whereas each task’s corresponding decoder needs to learn task-specific features to avoid underfitting. Mask R-CNN [19] adds an object mask prediction branch on top of the Faster R-CNN network and performs parallel classification and object detection box regression branches on the region of interest. LSNet [20] effectively combines instance segmentation, object detection, and pose estimation into a single network and proposes a loss function called Cross-IoU. MultiNet [21] is an efficient feedforward architecture that can perform joint semantic segmentation, image classification, and object detection. This architecture uses a single encoder and three decoders trained separately for each of the three tasks.

In research on applying multi-task learning networks to autonomous driving scenarios, YOLOP is an excellent multi-task learning network that performs well in three practical tasks, namely, vehicle detection, drivable area segmentation, and lane segmentation, and achieves outstanding performance on the BDD100K dataset. Furthermore, HyBridNet [22] improves upon YOLOP by using EfficientNet [23] as the feature extraction backbone and introduces bidirectional feature pyramids (BiFPN) for feature fusion at different semantic levels, resulting in significant improvements in recall rate for traffic object detection and accuracy for lane segmentation. In addition, some multi-task networks [24] also adopt the design principles of YOLOP and HyBridNet while incorporating attention mechanisms for network enhancement. The YOLOPv2 [25] network, which builds upon YOLOP, achieves superior performance by incorporating a refined feature extraction backbone derived from the YOLOv7 network. It exhibits cutting-edge performance in three crucial tasks: vehicle detection, drivable area segmentation, and lane line segmentation.

## 3. Methodology

Building upon the YOLOP network, this paper proposes a more efficient multi-tasking network architecture—YOLOP-DN, as depicted in Figure 2. Our model employs an improved multi-level feature extraction network as the backbone, which enhances performance while controlling the number of model parameters and computational complexity. To address the limitations of the YOLOP network in segmentation tasks, we design a novel dual-neck structure that is dedicated to detection and segmentation tasks separately, catering to the distinct feature requirements of each task. Furthermore, we design different decoders for the three distinct tasks. Our experiments demonstrate that this new architecture effectively improves the overall performance of the network. In this section, we will provide a detailed overview of the network’s encoder, decoder, and loss functions.

### 3.1. Encoder

The encoder primarily consists of a backbone and neck structure.

#### 3.1.1. Backbone

The backbone of YOLOP-DN consists of the Focus module, C2SPD module, and SPPF module, which form the feature extraction network. The feature extraction backbone utilizes the Focus [4] module and SPPF [4] module to improve inference speed and adjust output size. Additionally, we propose a C2SPD module for feature extraction, which not only has strong feature extraction capabilities but also reduces information loss during the process. Its structure is shown in Figure 3. The C2SPD module is composed of the SPD-Conv module [26], C2F module, and Conv module. The C2F module combines the design principles of C3 module [27] and ELAN [28], considering the shortest gradient path and adopting parallel and residual structures, allowing the network to obtain richer gradient information while being lightweight. Compared to traditional stride convolution layers and pooling layers, the SPD-Conv module reduces information loss in stride convolution, thus improving the accuracy of vehicle detection. The Conv module consists of convolutional layers, normalization layers, and activation layers.

#### 3.1.2. Dual-Neck

In studying YOLOP and HybridNets, we found that when deep global features guide the learning of shallow local features, the vehicle detection task, drivable area segmentation task, and lane line segmentation task share a common neck structure. However, they do not complement each other’s information learning but instead affect the network’s learning of local information about lane lines and drivable areas. Therefore, our neck network adopts a dual-neck structure to mitigate the adverse effects between the detection and segmentation tasks. As shown in Figure 4, this structure consists of two parts.The first part consists of the CUCA module and BottleneckCSP [4] module, specifically designed for the vehicle detection task. The CUCA module is responsible for scale transformation, and the BottleneckCSP module is used to fuse shallow and deep features. The second part consists of the CUCA module, BottleneckCSP module, and CUCAU module, which are used for the drivable area segmentation task and lane line segmentation task. The CUCA module and CUCAU module are employed for scale transformation, and the BottleneckCSP module is responsible for the fusion of shallow and deep features. This dual-neck structure employs a parallel architecture, which helps alleviate the impact of increasing parameter count on the network’s inference speed. Additionally, it integrates the shallow features P2 and P3 with deep features from the feature extraction network, thereby enhancing the overall performance of the network.

### 3.2. Decoders

As vehicle detection, drivable area segmentation, and lane line segmentation tasks have different demands for feature information, our network has a separate decoder designed for each task, as shown in Figure 5.

For the vehicle detection task, we have employed an anchor-based multi-scale detection scheme similar to YOLOv4. To extract the position information of the target more effectively, we utilize a bottom-up Path Aggregation Network (PAN [29]) in the decoder.The output feature maps from PAN are then concatenated with the output feature maps P5 and P6 from the dual-neck structure, achieving a deep fusion of high-level semantic information and local position information. The detection is performed on this multi-scale fusion feature map. In order to reduce information loss, we have also introduced a BCSPD module in PAN, which is mainly composed of BottleneckCSP modules and SPD-Conv modules.

For the segmentation tasks, we employ two distinct decoders. The decoder for drivable area segmentation consists of the BCCU module, BCCDC module, and convolution module. The BCCU module and BCCDC module perform scale transformation and complement the information and refine the edges of the drivable area through Upsample, BottleneckCSP module, and deconvolution. The decoder receives feature maps from the dual-neck structure and applies adjacent interpolation upsampling layers and deconvolution layers for pixel-wise semantic segmentation of the obtained feature maps. This process generates the final feature map for drivable area segmentation, which is then used for pixel-wise semantic segmentation. The decoder for lane line segmentation consists of the BCCDC module and convolution module. The BCCDC module performs scale transformation and complements and refines the information of the lane lines using the BottleneckCSP module and deconvolution. Additionally, to meet the requirement of lane line segmentation for local detailed features, the decoder cascades the shallow feature map P1 from the feature extraction backbone network before utilizing the output feature maps from the dual-neck structure. The concatenated shallow feature map is further supplemented with detailed information through two deconvolution layers. Finally, the obtained feature map is used for pixel-wise semantic segmentation in the lane line segmentation task.

### 3.3. Flowchart

The YOLOP-DN network is a multi-task learning network inspired by networks such as YOLOP, HybridNet, and YOLOPv2. Building upon the YOLOP network architecture, this network incorporates an efficient feature extraction backbone and introduces a dual-neck structure. It also designs three specific decoders for three different tasks. The flowchart of the network is depicted in Figure 6.

### 3.4. Loss Function

For the three tasks of vehicle detection, drivable area segmentation, and lane segmentation, we divided the loss function of the network into three parts.

Initially, it should be noted that the detection loss function denoted as $L_{\det}$, employed for the vehicle detection task, encompasses three distinct losses, namely, category loss $L_{\mathrm{class}}$, confidence loss $L_{\mathrm{obj}}$, and regression loss $L_{\mathrm{box}}$, as exemplified in Equation (Equation 1).
(1)$$L_{det}=\alpha_{1}L_{\mathrm{dass}}+\alpha_{2}L_{\mathrm{obj}}+\alpha_{3}L_{\mathrm{box}}.$$

It is worth noting that, to address the issue of sample imbalance, we have adopted the Focal loss [30] for the category loss and confidence loss. Additionally, we have employed the CIoU loss for the regression loss to measure the differences in overlap, aspect ratio, and scale between the predicted results and the ground truth. The respective weights for each of the component losses are denoted by $\alpha_{1}$, $\alpha_{2}$, and $\alpha_{3}$.

Furthermore, to tackle the issues of sample distribution imbalance, sparsity, and fragmented lane lines in the drivable area segmentation and lane line segmentation tasks, we have devised a hybrid loss that combines Tversky loss [31] with focal loss, as outlined in Equation (Equation 2).
(2)$$L_{seg}=L_{Tversky}+\lambda L_{Focal}.$$

It should be noted that component $L_{\mathrm{Tversky}}$ is predominantly leveraged to alleviate the issue of sample distribution imbalance, thus enabling the model to strike a more optimal balance between precision and recall. Conversely, component $L_{\mathrm{Focal}}$ is primarily deployed to facilitate the model’s ability to concentrate on learning from challenging samples, thereby augmenting the overall performance of the model. The definitions of the two distinct types of losses, namely loss $L_{\mathrm{Tversky}}$ and loss $L_{\mathrm{Focal}}$, are, respectively, specified in Equations (Equation 3) and (Equation 4).
(3)$$L_{Tversky}=1-\frac{TP_{p}}{TP_{p}+\varphi FN_{p}+\left(1-\varphi \right)FP_{p}}.$$
(4)$$L_{Focal}=-\lambda \frac{1}{N}\sum_{n=1}^{N}g_{n}{\left(1-p_{n}\right)}^{r}\log\left(p_{n}\right).$$

To recapitulate, our complete loss function is the weighted summation of three distinct segments, namely, the traffic object detection loss, the drivable area segmentation loss, and the lane line segmentation loss, as exemplified in Equation (Equation 5).
(5)$$L_{all}=\hat{\beta_{1}L_{det}}+\beta_{2}L_{da-seg}+\beta_{3}L_{ll-seg}.$$

In this context, the parameters $\beta_{1}$, $\beta_{2}$, and $\beta_{3}$ signify the individual weights attributed to each element of the loss function, whereas $L_{\mathrm{da}-\mathrm{seg}}$ and $L_{\mathrm{ll}-\mathrm{seg}}$ signify the losses incurred from the drivable area segmentation and lane line segmentation, respectively, as explicated in Equation (Equation 2).

## 4. Experiments

In this section, we will provide details on the experimental dataset, parameter configuration, and the obtained results. All the experiments described in this paper were carried out using an environment equipped with an NVIDIA GeForce RTX 3060 graphics card and the torch 1.13.0+cu117 learning framework.

### 4.1. Dataset

For all experiments, we utilized the BDD100K dataset as our experimental dataset. This dataset supports multi-task learning research in the field of autonomous driving, with annotations for 10 tasks and 100,000 frames of driving videos, making it the largest driving video dataset currently available. The BDD100K dataset covers a wide range of weather conditions and time periods. It includes images captured in sunny, cloudy, and rainy weather, as well as different time periods such as daytime and nighttime. This diversity allows the dataset to better reflect the variety and challenges present in real-world driving scenarios. By training and evaluating models using such a diverse dataset, it enhances their robustness and adaptability. Models become capable of making accurate predictions and decisions under various weather and lighting conditions. This comprehensive training helps in preparing the models to handle the complexities and uncertainties that arise in different driving scenarios. We used 70,000 images from the training set and 10,000 images from the validation set of the BDD100K dataset for the training and validation of our model. Moreover, as the labels for the 20,000 images in the test set are not publicly available, we evaluated the performance of our model on the validation set.

### 4.2. Training Protocol

During the training process, we employed the Adam optimizer with an initial learning rate of 0.01 and adopted a “cosine annealing” strategy to adjust the learning rate during training, with warm-up [32] training in the first three epochs. Additionally, momentum decay and weight decay were set to 0.937 and 0.005, respectively. The batch size for training was set to 16, and the total number of training epochs was set to 200. The size of the images was reduced from 1280 × 720 × 3 to 640 × 640 × 3 during the training stage, and from 1280 × 720 × 3 to 640× 384 × 3 during the testing stage. In the hyperparameter settings for the loss function, we set the hyperparameters for the detection loss as $\alpha_{1}=0.5$, $\alpha_{2}=1$, and $\alpha_{3}=0.05$, and the hyperparameter for the segmentation loss was set as λ=1. For the total loss hyperparameter settings, we set $L_{\det}$, $L_{\mathrm{da}-\mathrm{seg}}$ and $L_{\mathrm{ll}-\mathrm{seg}}$ as $\beta_{1}=0.2$, $\beta_{2}=0.2$, and $\beta_{3}=0.2$, respectively.

### 4.3. Experimental Results

We conducted qualitative and quantitative comparisons of our proposed model with a set of existing studies, primarily focusing on the performance of the model in each task, the number of parameters, and inference speed. In addition, we designed corresponding ablation experiments to evaluate the effectiveness of our work. In these experiments, we evaluate the performance of our proposed network using evaluation metrics commonly employed by state-of-the-art networks. For the detection task, we use mean average precision at IoU threshold of 0.50 (mAP50) and recall rate to assess the model’s detection performance. In autonomous driving scenarios, missing vehicle detections can have severe consequences, making recall rate an important metric for evaluating the model’s effectiveness. For the drivable area segmentation task, we utilize mean intersection over union (mIOU) to evaluate the accuracy of the segmentation. The mIOU value measures the overlap between the predicted drivable area and the ground truth, providing an assessment of the segmentation quality. For the lane line segmentation task, we employ accuracy and intersection over union (IOU) as evaluation metrics. Accuracy measures the overall correctness of the lane line predictions, whereas IOU evaluates the overlap between the predicted lane lines and the ground truth. These metrics provide insights into the model’s performance, considering the slender nature of lane lines in autonomous driving scenarios.

#### 4.3.1. Vehicle Detection Results

The experimental results of vehicle detection are presented in Table 1, where mAP50 and recall are used as evaluation metrics, and the top three performance indicators are highlighted in bold. As shown in the table, our model achieves the highest mAP50 of 78.1% and the second-highest recall of 90.5%, with an improvement of 1.6% and 1.3% in mAP50 and recall, respectively, compared to the baseline model.

#### 4.3.2. Drivable Area Segment Results

The results of the drivable area segmentation experiment are shown in Table 2, where mIoU is used as the evaluation metric, and the top three performance metrics are highlighted in bold. As shown in the table, our model achieved the highest mIoU of 92.0%, with a 0.5% improvement over the baseline model.

#### 4.3.3. Lane Segment Results

In the BDD100K dataset, lane markings are annotated with two lines, requiring preprocessing. First, we calculate the center line based on the two annotated lines, then draw an 8-pixel-wide lane mask for training, while keeping the lane line width at 2 pixels for testing. The results of the lane segment experiment are shown in Table 3, where accuracy and IoU are used as evaluation metrics, and the top three performance metrics are bold. As shown in the table, our model achieved a second-highest accuracy of 73.8% and a second-highest IoU of 27.3%, with significant improvements of 3.3% and 1.1%, respectively, over the baseline model.

#### 4.3.4. Model Parameter and Inference Speed

Table 4 presents a comparison among YOLOP, HybridNets, YOLOv2, and our model. All tests were conducted under the same experimental settings and evaluation metrics. The results demonstrate that the number of parameters and the inference speed of our model are relatively moderate.

#### 4.3.5. Discussion of Visualizations

Figure 6 depicts the visual comparisons of YOLOP, Hybridnet, and our proposed model on the BDD100K dataset. Figure 7 presents the results during the daytime.

The left column shows four scenarios of YOLOP, in which the first scenario has false drivable segments and disconnected lane predictions, the second scenario contains incorrect detection boxes, false drivable regions, and disconnected lane predictions, the third scenario has false drivable regions and missing detections on lanes, and the fourth scenario suffers from missed detections, false drivable regions, and disconnected lane predictions. The middle column displays the four scenarios of Hybridnet, in which the first scenario has incorrect detection and missing drivable region segmentation, the second scenario exhibits duplicated detections of small vehicles, missed vehicle detections, and disconnected lane detections, the third scenario has missed vehicle detections and false drivable regions, and the fourth scenario has missed vehicle detections, false drivable regions, and missing lane predictions. The right column shows the results of our proposed YOLOP-DN model, which achieves better performance in various daytime scenarios.

Figure 8 showcases the results obtained during nighttime scenarios. The left column displays four different scenarios using YOLOP, where the first scenario presents false drivable areas and lane prediction misses, the second scenario displays redundant vehicle detections and false drivable areas, the third scenario exhibits incorrect drivable areas, and the fourth scenario shows redundant detections, misses, and incorrect drivable areas. The middle column showcases Hybridnet’s results, where the first scenario displays lane prediction misses and incorrect detection boxes, the second scenario presents false drivable areas and false positives, the third scenario exhibits false drivable areas, missed drivable areas, and redundant vehicle detection boxes, and the fourth scenario showcases misses and disconnected lane predictions. The right column presents the results obtained using YOLOP-DN, which outperforms the other models in all nighttime scenarios.

#### 4.3.6. Qualitative and Quantitative Results Analysis

From a quantitative perspective, we can categorize the networks that simultaneously accomplish three practical tasks—vehicle detection, drivable area segmentation, and lane line segmentation—into two groups. The first group consists of networks with fewer parameters, such as YOLOP, Hybridnet, and YOLOP-DN. The second group includes networks with a larger number of parameters, such as YOLOv2. First, Hybridnet improves upon the YOLOP network by increasing the computational complexity, leading to significant improvements in lane line segmentation and detection recall. However, the inference speed of the network is greatly affected. Second, YOLOPv2 enhances feature extraction by incorporating an improved feature extraction backbone based on the YOLOv7 network. It achieves state-of-the-art performance in terms of accuracy and speed. However, the parameter count of YOLOPv2 is nearly five times that of YOLOP. Finally, YOLOP-DN introduces a dual-neck structure and three different decoders while minimizing the increase in parameter count and inference time. It achieves good performance in vehicle detection and drivable area segmentation tasks. However, there is a performance gap in lane line segmentation compared to the best-performing networks, although there is some improvement compared to YOLOP.

From a qualitative perspective, focusing on networks with fewer parameters, such as YOLOP, Hybridnet, and YOLOP-DN, we can analyze their performance. Visual results from Figure 7 and Figure 8 demonstrate that, regardless of daytime or nighttime scenarios, YOLOP-DN consistently outperforms both YOLOP and Hybridnet in terms of performance.

#### 4.3.7. Ablation Studies

Compared to the YOLOP network, our network has undergone many changes and improvements, and corresponding ablation experiments have been conducted. Table 5 shows the performance improvement of our model with these modifications, and the best performance indicators are highlighted in bold. The table shows that the introduction of the dual-neck structure relative to the baseline model resulted in a 1% improvement in the accuracy and IoU of lane segmentation. This innovation in the neck structure achieved improved segmentation performance while ensuring detection performance. The addition of the C2F module and SPD-conv module-based feature extraction backbone on top of the dual-neck structure resulted in a 1.4% and 1.3% improvement in mAP50 and recall for vehicle detection, respectively, a 0.5% improvement in mIoU for drivable area segmentation, and a 2.3% improvement in accuracy for lane segmentation compared to the baseline model. The improvement in the feature extraction backbone significantly improved the accuracy of both the detection and segmentation tasks without excessively increasing the number of parameters.

### 4.4. Application Deployment

In addition to the previous experiments, we also conducted deployment testing on the Atlas hardware to observe its actual performance.

#### 4.4.1. Hardware Configuration

The detailed hardware configuration of the Atlas 200I A2 device we used is as follows:

System: Ubuntu 22.04; RAM: LPDDR4X, 4 GB (an additional PCIe 3.0 M.2 SSD is installed, with 4GB allocated as swap memory); storage: MicroSD card with a maximum transfer speed of 140 MB/s, 64 GB; CPU: Ascend 310 series AI processor; 1 DavinciV300 AI core (500 MHz); 4 TAISHANV200M processor cores (1GHz); computing power: 8 TOPS INT8; 4 TFLOPS FP16.

#### 4.4.2. Deployment Result

After applying FP16 quantization to the neural network model and setting the semantic segmentation output resolution to 320 × 320, we performed inference on a H.264 video file with a resolution of 1280 × 720. The device underwent video decoding and neural network inference, with the results visualized in a Jupyter Notebook.

When using the OpenCV framework for inference, the memory utilization reached 82%, CPU utilization reached 99%, and the NPU utilization was 0%. The actual frame rate achieved was around 6 frames per second. When using the CANN framework for inference, the memory utilization reached 77%, CPU utilization reached 95%, and the NPU utilization was 99%. The actual frame rate achieved was around 18 frames per second.

## 5. Conclusions

This paper presents an efficient end-to-end multi-task learning network, YOLOP-DN, for simultaneously performing three driving perception tasks: vehicle detection, drivable area segmentation, and lane detection. The network consists of an efficient feature extraction backbone, a novel dual-neck structure, and three distinct functional heads. Without significantly increasing the number of parameters or decreasing inference speed, the network achieves good performance on the challenging BDD100K dataset. It is the first to realize real-time reasoning on embedded device Atlas 200I A2, which ensures that our network can be used in real-world scenarios. Furthermore, we validate the effectiveness of our model improvements through corresponding ablation experiments.

In future work, we plan to focus on two main research aspects. First, we will pay attention to the robustness of the model in real-world scenarios, especially in challenging environments such as heavy rain, fog, and low-light conditions. These environmental conditions can impact the accuracy of the model. To address these issues, we will explore the introduction of depth estimation as an auxiliary judgment or splitting the network into two stages, with the first stage performing image de-raining or de-fogging, and the second stage performing multi-task inference. These strategies may affect the inference speed of the model, so we plan to further research network architecture, network reparameterization, and network lightweighting methods. The other aspect is to focus on network performance research. By conducting inference on the test set and evaluating the model’s performance in actual deployment, there is still room for improvement in terms of accuracy and real-time performance. In terms of accuracy, we plan to research more efficient feature extraction modules, such as attention mechanisms. For real-time performance, we plan to explore more efficient parallel inference strategies. Additionally, we will explore the applicability of the model in broader domains, such as deploying the network in water surface cleaning boats or water surface search drones for tasks such as water surface debris detection, water surface vegetation segmentation, and water surface segmentation. These research directions will contribute to improving the robustness, accuracy, and real-time performance of the model and applying it to a wider range of application areas such as water surface cleaning and search and rescue.

## Figures and Tables

**Figure 1 sensors-24-01547-f001:**
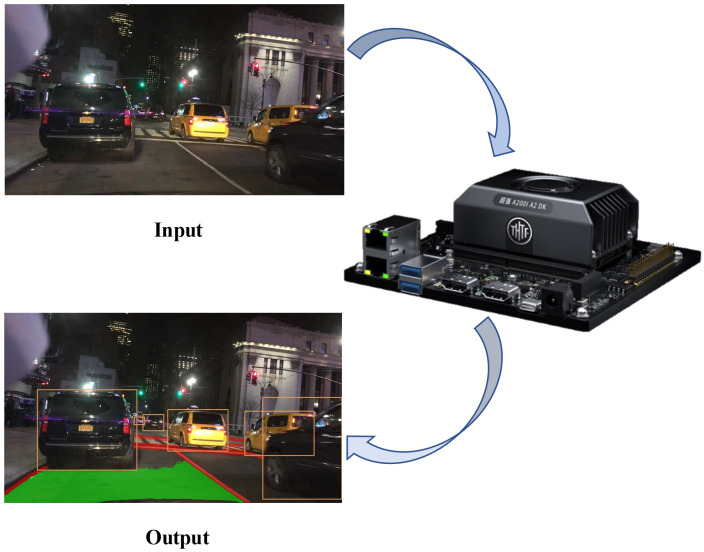
Performing real-time inference on Atlas 200I A2.

**Figure 2 sensors-24-01547-f002:**
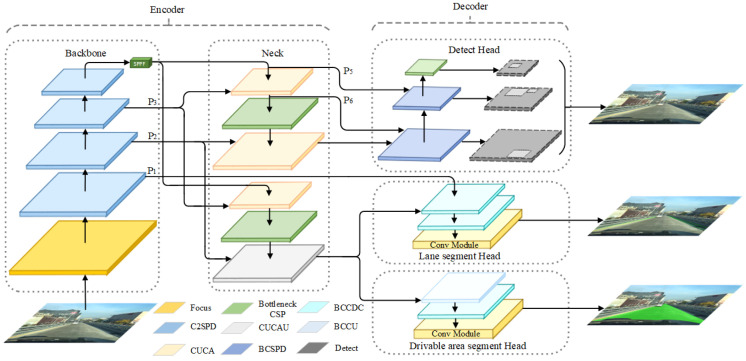
The network of YOLOP-DN.

**Figure 3 sensors-24-01547-f003:**
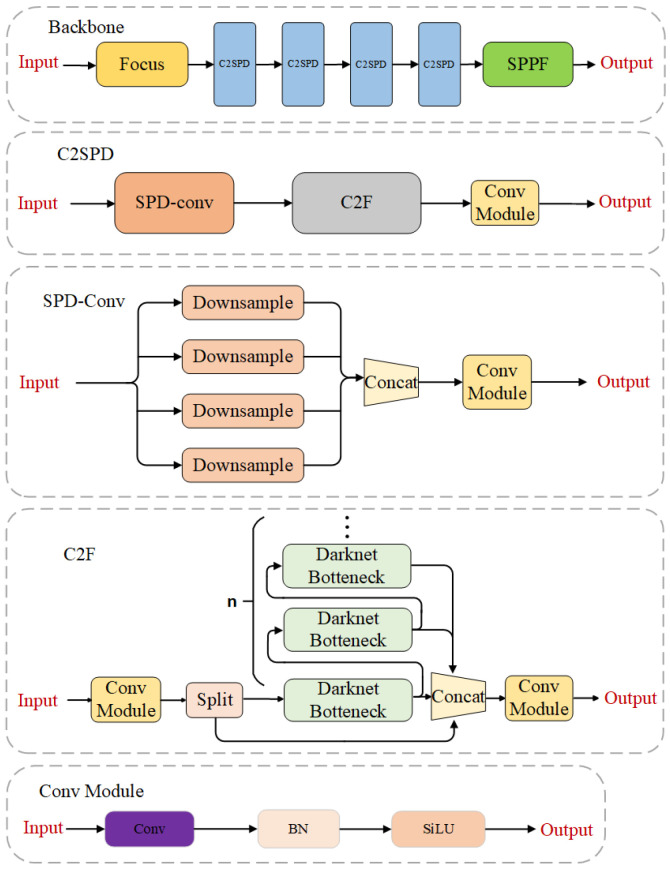
The backbone of feature extraction.

**Figure 4 sensors-24-01547-f004:**
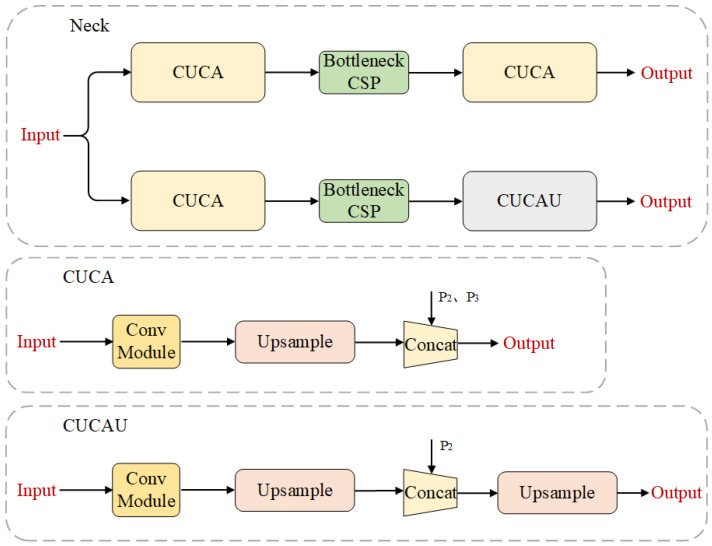
Dual-neck structure.

**Figure 5 sensors-24-01547-f005:**
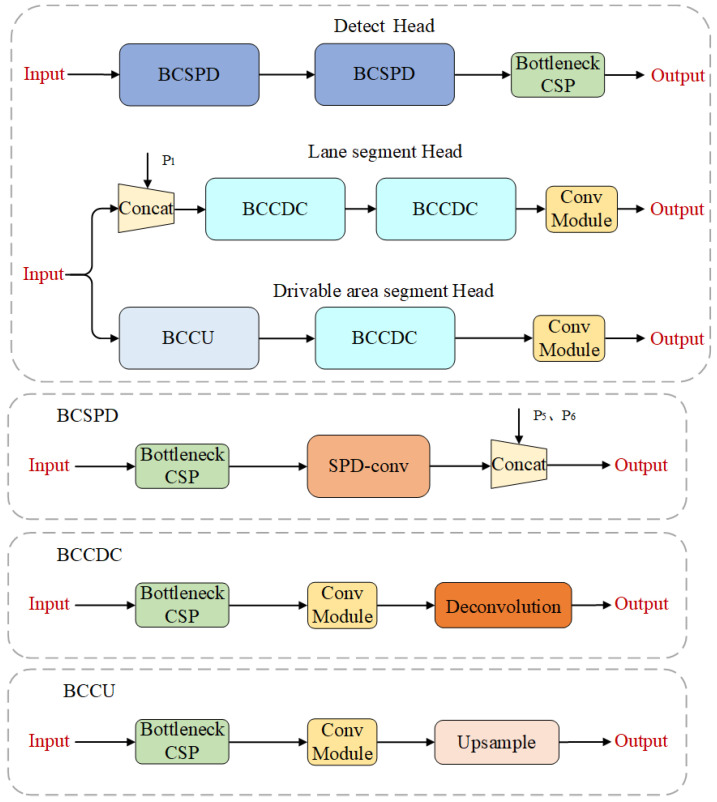
Decoders structure.

**Figure 6 sensors-24-01547-f006:**
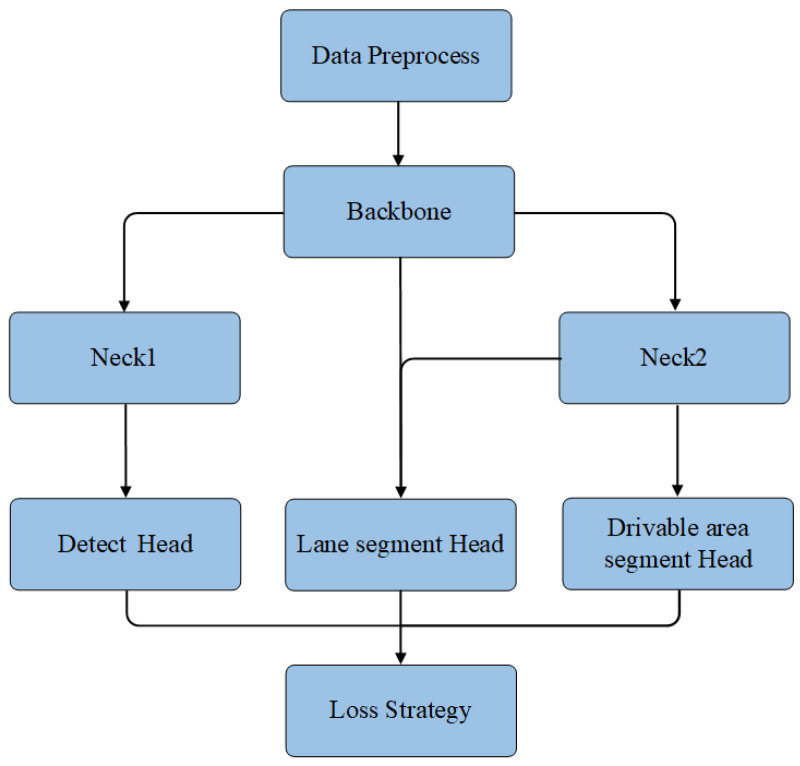
Methodology flow chart.

**Figure 7 sensors-24-01547-f007:**
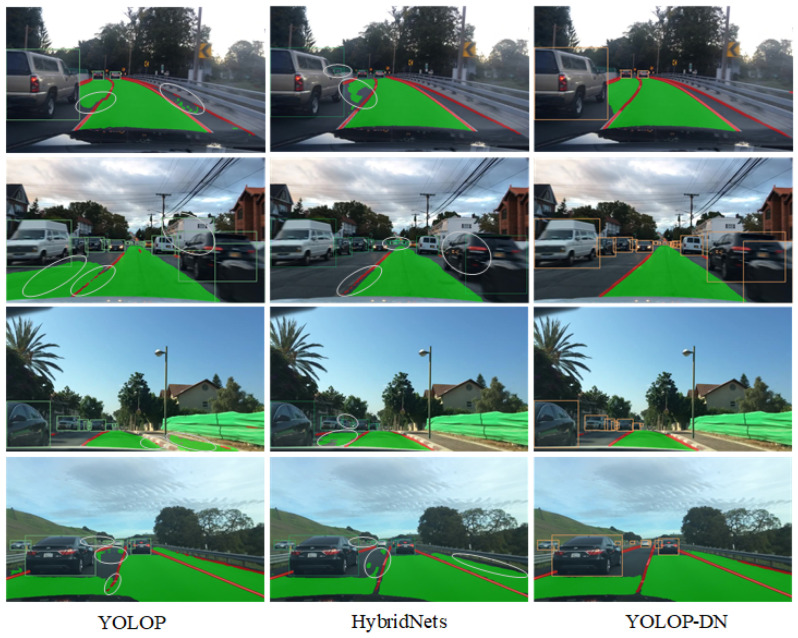
The daytime results.

**Figure 8 sensors-24-01547-f008:**
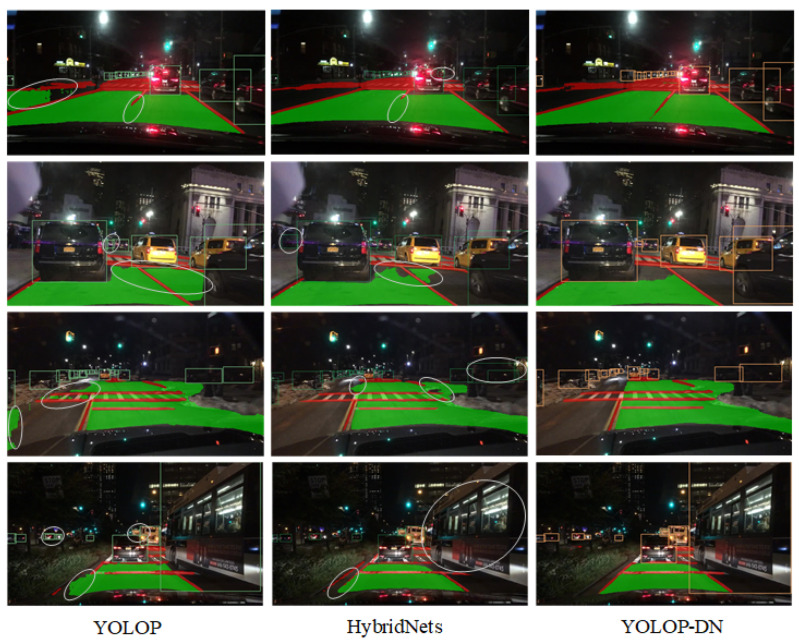
The nighttime results.

**Table 1 sensors-24-01547-t001:** Results on vehicle detection.

Network	mAP50 (%)	Recall (%)
MultiNet	60.2	81.3
DLT-Net [33]	68.4	89.4
Faster R-CNN	55.6	77.2
YOLOv5s	77.2	86.8
YOLOP (Baseline)	76.5	89.2
HybridNets	**77.3**	**92.8**
YOLOP-DN (Ours)	**78.1**	**90.5**
YOLOPv2 (SOTA)	**83.4**	**91.1**

**Table 2 sensors-24-01547-t002:** Results on drivable area segment.

Network	Drivable mIoU (%)
MultiNet	71.6
DLT-Net	71.3
PSPNet	89.6
YOLOP (Baseline)	**91.5**
HybridNets	90.5
YOLOP-DN (Ours)	**92.0**
YOLOPv2 (SOTA)	**93.2**

**Table 3 sensors-24-01547-t003:** Results on lane segment.

Network	Accuracy (%)	Lane IoU (%)
ENet [34]	34.12	14.64
SCNN	35.79	15.84
ENet-SAD	36.56	16.02
YOLOP (Baseline)	70.50	26.20
HybridNets	**85.40**	**31.60**
YOLOP-DN (Ours)	**73.80**	**27.30**
YOLOPv2 (SOTA)	**87.31**	**27.25**

**Table 4 sensors-24-01547-t004:** Comparison of model parameter and inference speed.

Network	Size (Pixel)	Params (M)	Speed (fps)
YOLOP (Baseline)	640	7.9	125
HybridNets	640	12.8	25
YOLOP-DN (Ours)	640	10.9	91
YOLOPv2 (SOTA)	640	38.9	168

**Table 5 sensors-24-01547-t005:** Evaluation of efficient experiments.

Architectures	Speed (fps)	mAP50 (%)	Recall (%)	mIoU (%)	Accuracy (%)	IoU (%)
YOLOP (Baseline)	**125**	76.5	89.2	91.5	70.5	26.2
+Double-Neck	115	76.7	89.2	91.7	71.5	27.2
+C2f	100	77.5	90.3	92.0	**74.2**	27.0
+SPD-conv	91	**78.1**	**90.5**	**92.0**	73.8	**27.3**

## Data Availability

Publicly available datasets were analyzed in this study. The BDD100K dataset can be found at https://bdd-data.berkeley.edu/.

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
