# Peer review of "A Multi-Task Network Based on Dual-Neck Structure for Autonomous Driving Perception"

_sensors, 2024, doi:10.3390/s24051547_

Round 1

Reviewer 1 Report

Comments and Suggestions for Authors

Dear authors,

I commend your extensive research and simulation efforts on the novel structure for real-time, three-task autonomous driving perception. However, I propose two enhancements that could potentially augment the impact of your publication.

1. In Section 3, which discusses the methodology, I suggest a more detailed textual description for each element in Figures 3, 4, and 5. This would aid readers, particularly those unfamiliar with this field, in comprehending the architecture more thoroughly without the need to refer to external sources. For instance, in Subsection 3.1, not only should high-level modules such as “Focus” and “SPFF” be explained in the subfigures, but fundamental elements like the “Conv Module” should also be mentioned in the main text.

2. To facilitate the replication of your work, especially if the code will not be open-sourced, I recommend providing a more comprehensive explanation of the selection process for the hyperparameters of each neural network component, as well as the loss function.

I hope these suggestions prove useful in enhancing the clarity and reproducibility of your work. Looking forward to your revised manuscript.

Reviewer 2 Report

Comments and Suggestions for Authors

The submitted paper presents “A Multitask Network Based on Dual-Neck Structure for Autonomous Driving Perception Although the paper might have some novelties, some points need clarification:

 ·         The manuscript provides a detailed description of the proposed YOLOP-DN multi-task learning network for autonomous driving perception. However, it should include a more in-depth discussion of the specific functions and interactions of the introduced C2SPD feature extraction module and the dual-neck architecture to enhance clarity. Additionally, authors should provide visual aids, such as architectural diagrams or flowcharts, that could help readers better understand the proposed model's components and their relationships.

·         To enhance the depth of analysis, it would be valuable to discuss the choice of evaluation metrics for each task and compare them more comprehensively against other state-of-the-art models. This could involve a deeper exploration of the model's strengths and limitations compared to existing approaches.

·         The manuscript briefly mentions the real-time deployment of the proposed model on an embedded device (Atlas 200I A2). However, the authors should provide additional details on the inference speed achieved on the embedded device. Furthermore, discussing the model's resource efficiency regarding memory utilization and computational requirements would contribute to a more holistic assessment of its practical applicability.

·         While the BDD100K dataset is used for evaluation, the authors can discuss the dataset's characteristics and potential biases, addressing how well the model may generalize to diverse real-world driving scenarios. Additionally, insights into the model's performance in handling variations in lighting conditions, weather, and geographical locations would provide a more comprehensive understanding of its robustness.

·         The conclusion briefly mentions future work involving additional tasks such as depth estimation and addressing real-world challenges like rain and fog. Expanding on these points to outline specific research directions, challenges anticipated, and potential methodologies for addressing them would enhance the manuscript's contribution to the field. Additionally, authors can emphasize the broader implications of the proposed model beyond the current study.

Round 2

Reviewer 2 Report

Comments and Suggestions for Authors

The authors had made all the requested revisions. The work can be accepted as it is.